# Pharmacogenomics-assisted schizophrenia management: A hybrid type 2 effectiveness-implementation study protocol to compare the clinical utility, cost-effectiveness, and barriers

**Aniruddha Basu**[1‡], **Atanu Kumar Dutta**[2‡], **Bhavani Shankara Bagepally**[3‡], **Saibal Das**[4,5‡*], **Jerin Jose Cherian**[5,6], **Sudipto Roy**[6], **Pawan Kumar Maurya**[4], **Indranil Saha**[4], **Deepasree Sukumaran**[7], **Kumari Rina**[1], **Sucharita Mandal**[1], **Sukanto Sarkar**[1], **Manoj Kalita**[4], **Kalyan Bhowmik**[4], **Asim Saha**[4], **Amit Chakrabarti**[4]

1 Department of Psychiatry, All India Institute of Medical Sciences, Kalyani, India, 2 Department of Biochemistry, All India Institute of Medical Sciences, Kalyani, India, 3 Indian Council of Medical Research, National Institute of Epidemiology, Chennai, India, 4 Indian Council of Medical Research, Centre for Ageing and Mental Health, Kolkata, India, 5 Department of Global Public Health, Karolinska Institutet, Stockholm, Sweden, 6 Indian Council of Medical Research, New Delhi, India, 7 Department of Pharmacology, All India Institute of Medical Sciences, Kalyani, India

‡ These authors share first authorship on this work
* saibal.das@icmr.gov.in, saibal.das@ki.se

**Data Availability Statement:** This is a study protocol and does not contain any study data.

## Abstract

### Objectives

The response to antipsychotic therapy is highly variable. Pharmacogenomic (PGx) factors play a major role in deciding the effectiveness and safety of antipsychotic drugs. A hybrid type 2 effectiveness-implementation research will be conducted to evaluate the clinical utility (safety and efficacy), cost-effectiveness, and facilitators and barriers in implementing PGx-assisted management compared to standard of care in patients with schizophrenia attending a tertiary care hospital in eastern India.

### Methods

In part 1, a randomized controlled trial will be conducted. Adult patients with schizophrenia will be randomized (2: 1) to receive PGx-assisted treatment (drug and regimen selection depending on the results of single-nucleotide polymorphisms in genes *DRD2*, *HTR1A*, *HTR2C*, *ABCB1*, *CYP2D6*, *CYP3A5*, *and CYP1A2*) or the standard of care. Serum drug levels will be measured. The patients will be followed up for 12 weeks. The primary endpoint is the difference in the Udvalg for Kliniske Undersøgelser Side-Effect Rating Scale score between the two arms. In part 2, the cost-effectiveness of PGx-assisted treatment will be evaluated. In part 3, the facilitators and barriers to implementing PGx-assisted treatment for schizophrenia will be explored using a qualitative design.

**Funding:** This study is funded by ICMR, New Delhi, India through the ICMR-National Taskforce on Safe and Rational Use of Medicines (extra-mural fund). The funders had no role in study design, data collection and analysis, decision to publish, or preparation of the manuscript.

**Competing interests:** I have read the journal's policy and the authors of this manuscript have the following competing interests: This study is funded by ICMR, New Delhi, India (https://www.icmr.gov.in/) through the ICMR-National Taskforce on Safe and Rational Use of Medicines (extra-mural fund). The fund was awarded to SD (File No. SRUM/2023/BMS/Part2/SaibalDas). The funders had no role in study design, data collection and analysis, decision to publish, or preparation of the manuscript.

## Expected outcome

The study findings will help in understanding whether PGx-assisted management has a clinical utility, whether it is cost-effective, and what are the facilitators and barriers to implementing it in the management of schizophrenia.

## Trial registration

The study has been registered with the Clinical Trials Registry–India (CTRI/2023/08/056210).

## Introduction

The global burden of schizophrenia is 23.6 (95% confidence interval: 20.2–27.2) millions [1] and the prevalence of schizophrenia in India is 0.3% (95% confidence interval: 0.2–0.3%) [2]. The response to antipsychotic therapy is highly variable, and it is not possible to predict those patients who will or will not respond to drugs. Furthermore, around 30% of these patients are treatment-resistant [3]. Across several studies, it has been found that the proportion of patients receiving antipsychotic polypharmacy ranged from 15.9–60.5% before they received clozapine, the last pharmacotherapeutic resort [4]. Treatment of these patients imposes a huge burden on the patient, their caregivers, and the health system. The typical and atypical antipsychotics derive their therapeutic benefit predominantly from the antagonism of dopamine ($D_2$) and serotonin (5-$HT_{2A}$) receptors. Many of these compounds are associated with common and significant adverse effects [e.g. metabolic syndrome, hyperprolactinemia (in females) extrapyramidal symptoms, anticholinergic effects, sedation, etc.] [5] which negatively impact adherence. Pharmacogenomic (PGx) factors play a major role in deciding treatment responses to antipsychotic drugs [3].

Multiple studies have investigated PGx approaches to identify genotype-specific dosing and predict antipsychotic responses and/or adverse effects [6]. Currently, the United States Food and Drug Administration provides information on PGx biomarkers in their drug labeling of nine antipsychotic drugs [7]. Similarly, the Pharmacogenomics Knowledge Base website lists ten antipsychotics where caution is advised for patients who are poor cytochrome 2D6 (CYP2D6) metabolizers [8]. Drug labels with PGx information are provided for specific antipsychotics. The Royal Dutch Association for the Advancement of Pharmacy-Dutch Pharmacogenetics Working Group has provided PGx drug dosing guidelines based on CYP2D6 genotypes for six antipsychotics [9]. However, negative result was also been reported. In the Danish population, it was found that routine CYP testing does not affect the persistence of antipsychotic drug treatments [10].

Medication optimization interventions (e.g. PGx-assisted treatment) based on the concept of precision medicine in people on polypharmacy due to psychotic disorders are complex and limited; a more holistic and integrated approach is warranted [11,12]. Several observational studies have reported associations between genetic variants and treatment response; however, there are limited studies that have used PGx as an interventional tool in assisting the selection of antipsychotic drugs and their doses to optimize schizophrenia treatment. This is particularly important in the ethnically diverse Indian context as differential pharmacokinetics and pharmacodynamics of antipsychotic drugs can lead to a major impact on their effects. Further, it is also important to evaluate the cost-effectiveness of PGx-assisted treatment in schizophrenia in a lower-middle-income country like India. Finally, it is important to generate insights from the patients, caregivers, and psychiatrists on the facilitators and barriers to implementing PGx-

assisted treatment in schizophrenia. Hence, this hybrid type 2 effectiveness-implementation research will be conducted to evaluate the clinical utility (safety and efficacy), cost-effectiveness, and facilitators and barriers in implementing PGx-assisted management as compared to the standard of care in patients with schizophrenia. Hybrid type 2 effectiveness-implementation research combines elements of both effectiveness and implementation research within a single study. This approach aims to not only assess the effectiveness of an intervention or program in real-world settings but also focuses on understanding and enhancing the implementation process.

## Methods

The study duration will be three years and the three parts of the study will be performed chronologically. The study will be conducted at the Indian Council of Medical Research (ICMR)-Centre for Ageing and Mental Health, Kolkata, India, in collaboration with the Departments of Psychiatry, Biochemistry, and Pharmacology, All India Institutes of Medical Sciences, Kalyani, India; ICMR-National Institute of Epidemiology, Chennai, India; and ICMR Headquarters, New Delhi.

### Part 1

**Study design and setting.** This will be a randomized, parallel-arm, patient and assessor-blinded study. The SPIRIT (Standard Protocol Items: Recommendations for Interventional Trials) checklist has been followed (Fig 1, S1 File).

**Eligibility criteria.** Adult (age ≥18 years) patients of any gender residing in the eastern Indian region for at least three consecutive generations; attending the outpatient Department of Psychiatry, All India Institutes of Medical Sciences, Kalyani, India; diagnosed with schizophrenia (Diagnostic and Statistical Manual of Mental Disorders, fifth edition) for ≥6 months;

| | Study period | | | | | |
|---|---|---|---|---|---|---|
| | Enrolment | Allocation | Post-allocation | | | |
| Time point | -t | 0 | 2 weeks | 4 weeks | 8 weeks | 12 weeks |
| **Enrolment** | X | | | | | |
| Eligibility screen | X | | | | | |
| Informed consent | X | | | | | |
| Capacity assessment | X | | | | | |
| Allocation | | X | | | | |
| **Interventions** | | | | | | |
| Arm A: PGx-assisted treatment | | | ←————————————→ | | | |
| Arm B: Standard of care | | | ←————————————→ | | | |
| **Assessments** | | | | | | |
| Baseline variables* | X | X | | | | |
| UKU-SERS | | | X | X | X | X |
| PANSS | | X | X | X | X | X |
| CGI | | X | X | X | X | X |
| MoCA | | X | | | | X |
| Waist circumference and biochemistry† | | X | | | | X |
| Adverse and serious adverse events (SMARTS) | | | X | X | X | X |
| Hospitalization (if any) | | | X | X | X | X |
| Names and doses of various antipsychotic drugs | | X | X | X | X | X |
| Serum level of different antipsychotic drugs‡ | | X† | X† | X† | X† | X† |
| Drug adherence | | | X | | | X |
| EQ-5D-5L | | X | | | | X |

\* Socio-demographic characteristics (gender, age, and body mass index), lifestyle (diet, sleep, and exercise), addiction history (other than behavioral addiction), present medical history, present drug history, relevant family history of psychiatric and neurological diseases, and details of the present psychotic disorder (age of onset, duration, and the severity of present illness).
† Serum fasting glucose, lipid profile, and prolactin (for females).
‡ At steady state after initiation or change in dose (as applicable).

**Fig 1. SPIRIT (Standard Protocol Items: Recommendations for Interventional Trials) diagram for Part 1.** CGI, Clinical Global Impressions Scale; MoCA, Montreal Cognitive Assessment; PANSS, Positive and Negative Syndrome Scale; PGx, Pharmacogenomics; SMARTS, Systematic Monitoring of Adverse events Related to TreatmentS; UKU-SERS, Udvalg for Kliniske Undersøgelser Side-Effect Rating Scale.

Positive and Negative Syndrome Scale (PANSS) [13,14] total score of >60 with at least two positive items scored >4; those who are not previously genotyped; who will be adjudicated by the treating psychiatrist to be treated with one or more of the following first-line antipsychotic drugs: olanzapine, risperidone, haloperidol, amisulpride, quetiapine, aripiprazole, and trifluoperazine; will be recruited [15–17]. Patients having a history of treatment-resistant schizophrenia [3] and/or any past documented history of clozapine therapy, those who required electroconvulsive therapy over the past month or those who presently require urgent electroconvulsive therapy before initiating pharmacotherapy in the present study, those having a history of drug hypersensitivity or contraindication to the above-mentioned antipsychotic drugs, those with any organic mental health disorder, those with a well-documented history of epilepsy or hyperpyretic convulsion, those with severe violent or self-harm behavior that would limit the patient's ability to complete and/or participate in the study (based on investigator's judgment), those with severe unstable medical comorbidities (based on investigator's judgment), those who require long-acting injectable drug(s) to maintain treatment adherence, those with a history of neurolept malignant syndrome, those who are pregnant or breastfeeding at the time of enrolment, and those who will be unwilling to provide informed consent will be excluded (Table 1) [18]. Based on the existing patient load at the Department of Psychiatry, All India Institutes of Medical Sciences, Kalyani, India, the patient recruitment process is expected to be completed in 1.5 years.

**Genotyping.** Peripheral venous blood (3 ml) will be collected from patients of PGx arm (Arm A). Specific single-nucleotide polymorphisms (SNPs) [19,20] will be detected based on their role in the pharmacokinetics and pharmacodynamics of the mentioned antipsychotics and their prevalence (The 1000 Genomes Project [21] and Genome Asia [22] in the eastern Indian population (Table 2). Real-time TaqMan-based quantitative polymerase chain reaction (PCR) and Sanger sequencing will be performed [20]. Genomic DNA will be extracted from

**Table 1. Eligibility criteria of the study population.**

**Inclusion criteria**

- Adult (≥18 years) patients of any gender residing in the eastern Indian region for at least three consecutive generations and diagnosed with schizophrenia (DSM, fifth edition) for ≥6 months
- Positive and Negative Syndrome Scale (PANSS) total score of >60 with at least two positive items scored >4
- Not previously genotyped
- Adjudicated to be treated with one or more of olanzapine, risperidone, haloperidol, amisulpride, quetiapine, aripiprazole, and trifluoperazine

**Exclusion criteria**

- History of treatment-resistant schizophrenia and/or any past documented history of clozapine therapy
- History of electroconvulsive therapy over the past month or those who presently require urgent electroconvulsive therapy
- History of drug hypersensitivity or contraindication to the above-mentioned antipsychotic drugs
- Any organic mental health disorder
- A well-documented history of epilepsy or hyperpyretic convulsion
- Severe violent or self-harm behavior (investigator's judgment)
- Severe unstable medical comorbidities (investigator's judgment)
- Requirement of long-acting injectable drug(s) to maintain treatment adherence
- History of drug-induced malignant syndrome
- Pregnant or breastfeeding at the time of enrolment
- Unwilling to provide informed consent

DSM, Diagnostic and Statistical Manual of Mental Disorders.

**Table 2. Genes and respective single-nucleotide polymorphisms that will be detected by genotyping.**

| Gene | SNP | Gene | SNP |
|---|---|---|---|
| CYP2D6 | rs1058164 | ABCB1 | rs1045642 |
| | rs1065852 (CYP2D6*10) | | rs2032582 |
| | rs1135840 (CYP2D6*10) | | rs3842 |
| | rs16947 (CYP2D6*14) | CYP3A5 | rs776746 |
| | rs28371699 | DRD2 | rs1079597 |
| | rs28371702 | | rs2514218 |
| | rs28371725 (CYP2D6*41) | | rs4436578 |
| | rs3892097 (CYP2D6*4) | | rs6277 |
| HTR1A | rs10042486 | HTR2C | rs1283677 |
| | rs6313 | CYP1A2 | rs762551 |
| **Gene** | **SNP** | **Gene** | **SNP** |
| CYP2D6 | rs1058164 | ABCB1 | rs1045642 |
| | rs1065852 (CYP2D6*10) | | rs2032582 |
| | rs1135840 (CYP2D6*10) | | rs3842 |
| | rs16947 (CYP2D6*14) | CYP3A5 | rs776746 |
| | rs28371699 | DRD2 | rs1079597 |
| | rs28371702 | | rs2514218 |
| | rs28371725 (CYP2D6*41) | | rs4436578 |
| | rs3892097 (CYP2D6*4) | | rs6277 |
| HTR1A | rs10042486 | HTR2C | rs1283677 |
| | rs6313 | CYP1A2 | rs762551 |

SNP, single nucleotide polymorphism.

peripheral blood using a commercial kit as per the manufacturer's instructions. The respective gene loci will be amplified by using specific forward and reverse primers. The primer sequences will be checked for dimer and helix formation by using the Gene Runner® software. PCR will be carried from 100 ng genomic DNA in a total reaction volume of 20 μl. The reaction mixture will consist of 2 μl of 10X buffer, 0.3 μl of 10 mM dNTP, 0.3 μl of each 10 mM forward and reverse primers, and 0.2 μl of 5U/μl of Taq polymerase. Nuclease-free water will be added to make the final reaction volume to 20 μl. The amplified products will be checked on 1.5% agarose gel electrophoresis. Sanger sequencing and TaqMan-based genotyping would be carried out using specific protocols available from the kit manufacturers. The polymorphisms will be classified as wild type, homozygous, or heterozygous mutation by melting-curve analysis [20].

**Randomization, blinding, and allocation concealment.** Variable-sized block randomization will be performed based on computer-generated random numbers using open-access software (sealed envelope™) [23]. Allocation concealment will be ensured using serially numbered opaque sealed envelopes. The randomization code and the envelopes for allocation concealment will be prepared by the study statistician having no role in the participant enrolment process. The treating psychiatrists will be enrolling the trial participants and will assign them to the treatment arms as per the allocation sequence. The treating psychiatrists will be aware of the patient's treatment arm (unblinded). The patients and the outcome assessors will not be revealed the treatment arms and the genotyping results (patient and assessor blind) until the study completion (follow-up for 12 weeks).

**Intervention.** The recruited patients will be randomized (2: 1) into two arms as follows (Fig 2):

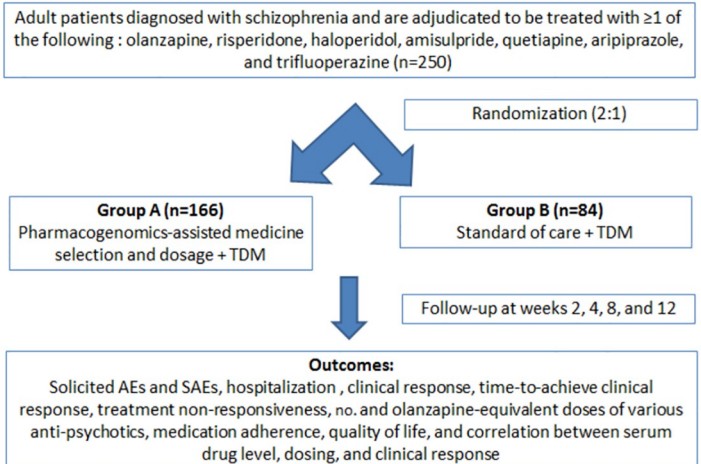

**Fig 2. Study flowchart.** TDM: Therapeutic drug monitoring.

- **Arm A:** PGx-assisted treatment (antipsychotic drug and regimen selection depending on the results of the mentioned SNPs).

- **Arm B:** Standard of care (as per standard practice).

Till the genotyping results come (7±2 days), the patients will be treated as per the standard treatment guidelines of the Indian Psychiatry Society. Based on the genotype results, the patients of Arms A will be classified as follows:

- **Poor metabolizers:** Those having two inactive alleles.

- **Intermediate metabolizers:** Those having decreased activity alleles or one active and one inactive allele or one decreased activity and one inactive allele.

- **Fast metabolizers:** Those having two functional alleles (wild type).

The standard dosages of the relevant antipsychotic drugs for first and multiple-episode schizophrenia are enlisted in Table 3 [24]. The antipsychotic drug selection and dosing algorithm [based on package inserts of United States Food and Drug Administration, European Medicines Agency, Health Canada, Pharmaceuticals and Medical Devices Agency, Japan; and professional PGx guidelines (Clinical Pharmacogenetics Implementation Consortium,

**Table 3. Standard dose range of antipsychotic drugs.**

| Antipsychotic drug | Usual daily dose (mg/day) | Maximum daily dose (mg/day) |
|---|---|---|
| Olanzapine | 10–30 | 30 |
| Risperidone | 2–8 | 16 |
| Haloperidol | 5–20 | 20 |
| Amisulpride | 50–800 | 1200 |
| Quetiapine | 300–800 | 800 |
| Aripiprazole | 10–30 | 30 |
| Trifluoperazine | 15–30 | 30 |

Pharmacogenomics Knowledge Base, and Royal Dutch Association for the Advancement of Pharmacy–Pharmacogenetics Working Group), and primary scientific literature] for initial treatment and follow-up treatment (including any change in drugs) are illustrated in Fig 3 [6,10,11,19,20]. The pharmacokinetics-related dosage adjustment will be based on the primary CYP450 metabolic enzymes. The antipsychotic drug and regimen in both arms may be altered based on the clinical response (efficacy or adverse events) as adjudicated by the treating psychiatrists during follow-up visits and the same will be recorded and used for the analysis. The algorithm will be updated during the study, if required, based on the findings of the interim analysis. The patients will be followed up at the end of 2, 4, 8, and 12 weeks.

**Therapeutic drug monitoring (TDM).** After reaching a steady state, TDM (measurement of serum drug levels) will be performed for each antipsychotic drug using trough samples. This will be performed by the high throughput ultra-performance liquid chromatographic method. A column size of 100 mm × 2.1 mm internal diameter and 1.7 m particle size will be used to separate the samples with gradient elution consisting of 0.1% trifluoroacetic acid and acetonitrile with a photodiode array detector. The elution time is expected to be 7 min with a 0.3 ml/min flow rate, 1 μl injection volume, and a 45˚C column oven temperature [25]. The Levey-Jennings chart will be used for quality assurance and quality control.

**Data collection and outcome measurements.** At the beginning, the functional capacity of the patients will be assessed as per the Mental Health Care Act, 2017 [26]. The socio-demographic characteristics (gender, age, and body mass index), lifestyle (diet, sleep, and exercise), addiction history (other than behavioral addiction), present medical history, present drug history, relevant family history of psychiatric and neurological diseases, and details of the present

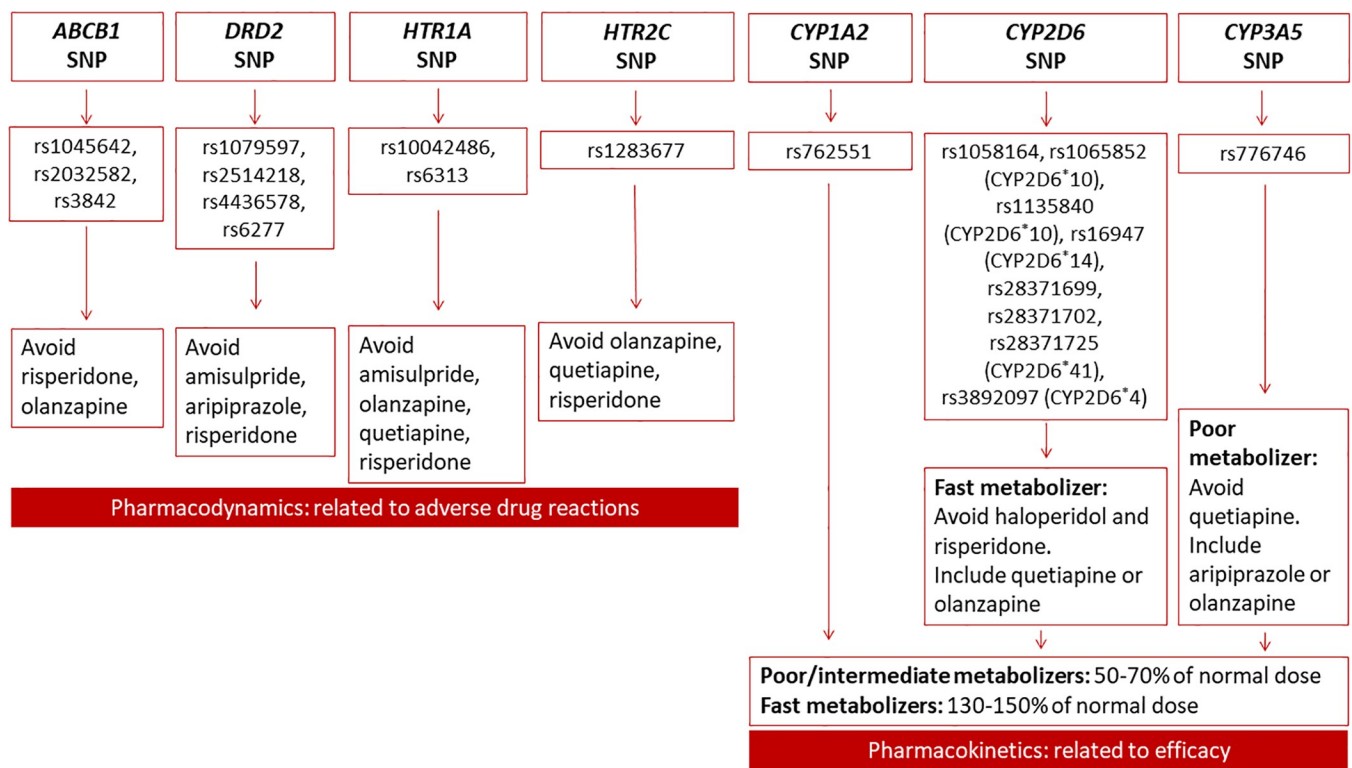

**Fig 3. Pharmacogenomics-assisted treatment algorithm.** rs, reference single nucleotide polymorphism; SNP, single nucleotide polymorphism.

psychotic disorder (age of onset, duration, and the severity of present illness) will be assessed at baseline. The primary endpoint will be the difference in the Udvalg for Kliniske Undersøgelser Side-Effect Rating Scale (UKU-SERS) [27] score between the two arms. The secondary endpoints will be clinical response at week 12 [PANSS [14,15] and Clinical Global Impressions Scale (CGI) scale [28]] scores; change in the cognitive function [Montreal Cognitive Assessment (MoCA) [29]; the proportion of patients developing adverse events and serious adverse events [Systematic Monitoring of Adverse events Related to Treatment S (SMARTS)] [30–32], including metabolic syndrome (based on waist circumference, serum fasting glucose, and lipid profile) [33], hyperprolactinemia (for females), anticholinergic effects, sedation, and extrapyramidal effects [5]; the proportion of patients requiring hospitalization; the duration of hospitalization; the time-to-achieve clinical response through 12 weeks; the proportion of patients unresponsive to treatment (<20% reduction in PANSS score at the end of six weeks as compared to the score at baseline) [34]; the number and olanzapine-equivalent doses of various antipsychotic drugs; the correlation between serum drug level, dosing, and clinical response; drug adherence (pill-counting method) [35]; and the quality of life (EQ-5D-5L) [36]. The scales selected are widely used and validated for determining the efficacy and safety outcomes of antipsychotic drugs. Fig 1 depicts the schedule of assessment of various outcome measurements.

**Sample size estimation.** Considering a difference of 0.62 [10] in the UKU-SERS score between the two arms, a standard deviation of 1.34 [10], a clinically meaningful difference of 1.18, a two-sided alpha error of 5%, a power of 90%, a randomization ratio of 2: 1, and a drop-out rate of 10%, the final estimated sample size will be 249 (round-up to 250) [166 in Arm A (PGx arm) and 83 in Arm B (standard of care arm)].

**Data management and statistical analyses.** The investigators will be responsible for assuring completeness, accuracy, and timely collection of data. Data will be entered in REDCap software (Vanderbilt, USA) by the study sites, and case record forms will be scanned in REDCap. The data will be checked for normal distribution (Kolmogorov-Smirnov test). The chi-squared test will be used for categorical variables (e.g., response rates, safety outcomes, etc.). For continuous variables (e.g. scale scores, duration of hospitalization), analysis of covariance with treatment in the model and baseline clinical and demographic characteristics and drug dosage as covariates will be used. Subgroup analysis will be performed based on the treatment status (treatment naïve and those who received prior treatment), socio-demographic characteristics, negative symptoms from the first psychotic episode, PANSS and CGI scores, comorbid substance use; age at onset, lack of early response, and treatment adherence.

The Kaplan-Meier estimate will be used to draw the survival curves denoting the time-to-achieve response. Cox's proportional hazards model will be used to assess the difference in the time to respond between the two arms allowing for other covariates. For violation in the proportional hazards assumption in Cox's proportional hazards model, violations will be identified through graphical methods and a sensitivity analysis will be performed to evaluate the robustness of the results to different modeling assumptions. This might involve using different cutoffs for time intervals or assessing the impact of different modeling strategies. If assumptions are not violated, based on the statistical criteria, we will choose an appropriate model.

For correlation between drug dosing, drug levels, and clinical response Pearson's or Spearman correlation test will be applied. Both intention-to-treat and per-protocol analyses will be performed. A p-value of <0.05 will be considered significant. Interim analyses will be performed after the follow-up of 80, 160, and 240 patients. The O'Brien-Fleming boundary will be used for stopping rules [37].

**Data monitoring.** The data monitoring will be performed by designated officials appointed by ICMR Headquarters, New Delhi.

## Part 2

**Study design.** This will be the cost-effectiveness component of the study.

**Cost-effectiveness analysis.** The intervention and outcome data of the same patients from Part 1 will be used. A decision analytic model will be developed from the health system's perspective from the patients' data using a short-term decision tree (Fig 4) and a long-term (lifetime horizon) Markov model. The simulation will begin with a study of individuals entering the model as the target population and proceeding into one of two treatment strategies. The Markov model structure will be developed as per the experience obtained from the trial as well as from a literature review. The trial findings will be used as transitional probabilities for the decision tree and the Markov model. The intervention costs will be estimated from the study using a testing panel including respective gene-drug pairs with 30 test-return days. The other relevant model input parameters, the utility values, the prevalence of schizophrenia, etc., will be retrieved from the literature and used in the models.

The model will estimate the health outcomes in terms of life years, quality-adjusted life years (QALYs), and costs. All future costs and consequences will be discounted at 3%. This implies that implies the application of a discount rate of 3% to account for the time value of money when considering future costs and consequences in the cost-effectiveness. The costs will be reported in 2023 Indian rupees and also in United States dollars. The effectiveness will be measured in terms of QALYs. The outcome will be the incremental cost-effectiveness ratio (ICER = $\Delta_{Cost}/\Delta_{QALY}$). One-way, scenario and probabilistic sensitivity analyses will be performed to assess the robustness of the findings due to variations in demographics, risk levels, and follow-up timeframe. The intervention will be considered cost-effective if the ICER is lower than the willingness to pay threshold, fixed based on the per capita gross domestic product as suggested by the World Health Organization [38]. A budget impact analysis will also be performed if the intervention is found to be cost-effective.

A budget impact analysis of the implementation of the PGx intervention strategy will be performed using the standard methods as per guidelines suggested by Health Technology Assessment India, Department of Health and Family Welfare, Government of India [39] only if it is found to be cost-effective. The annual budget impact will be estimated for a period of five years. The input costs for the budget impact analysis will be taken from the previously estimated model. The budget impact at the state level for each of the Indian states will be assessed considering the state-wise prevalence of schizophrenia and state-level eligible patient

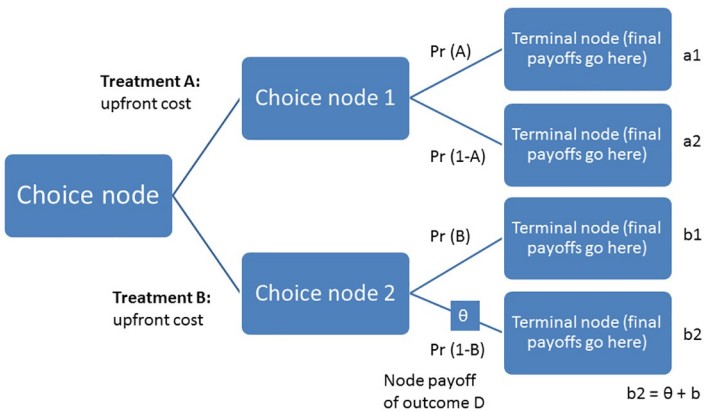

**Fig 4. Short-term decision tree model.** Pr: Probability.

population who will likely utilize PGx-assisted treatment if implemented at district-level hospitals. The budget required for offering the PGx-assisted treatment under the national program will be estimated using the following formula [40]:

$$B = N \times \left( C_{dt} + C_{My1} \right)$$

where B is the budget required for offering psychiatric treatment services to the eligible population, n is the eligible population estimated using a top-down approach, $C_{dt}$ is the unit cost of antipsychotic treatment from the decision tree Markov model (dt), and $C_{My1}$ is the cost of schizophrenia management for the first year as derived from the Markov model (M).

The estimated budget as the percentage increase will be calculated from the existing total healthcare budget for each of the respective states. No discount will be applied as the budget impact is the estimation of the financial cost. The health budget will be projected based on a 5% annual increase in health expenditure, while the estimated budget for schizophrenia will be projected using the population's annual growth rate until 2027. Further, the same method will be used to estimate the state-specific budget impact for the country-wide expansion of the treatment of schizophrenia.

## Part 3

**Study design.**  This will be the qualitative component of the study.

**Methodology.**  The facilitators and barriers to implementing PGx-assisted management in patients with schizophrenia will be explored. A narrative approach will be adopted. In-depth interviews and focused-group discussions of patients and their primary caregivers will be conducted. Key informant interviews will be conducted among psychiatrists. The interview guide will be prepared based on the following framework [41–46].

- General issues (facilitators and barriers) of the management of schizophrenia from the patient and familial perspectives.

- Impact of schizophrenia on individual, family, and societal well-being.

- Drug adherence and adverse reactions of antipsychotics.

- Hospitalization and the adverse consequences.

- The financial impact of management of schizophrenia.

- Expectations on precision medicine using PGx.

- Consideration of PGx-assisted management.

- Satisfaction with the quality of PGx testing and reports.

- Acceptability, feasibility, availability, and affordability of PGx-assisted management at different healthcare levels.

- Sustainability and scalability of PGx-assisted management.

- Facilitators and barriers in implementing PGx-assisted management.

**Sample size estimation and analysis.**  Patients, primary caregivers, and psychiatrists (six to twelve each) will be interviewed until saturation is obtained [47]. This sample size is tentative and the interview will continue till data saturation. Thematic saturation will be based on the three primary elements in its calculation and assessment: base size, run length, and new

information threshold [48]. A thematic analysis will be performed. This will involve a meticulous process of familiarization with the data, coding segments into meaningful units, and iteratively organizing these codes into coherent themes that encapsulate key concepts or patterns. Because of the flexible yet rigorous approach, thematic analysis will enable the exploration of diverse perspectives, facilitating the extraction of rich, contextually embedded insights that deepen our understanding of the facilitators and barriers to implementing PGx-assisted management in patients with schizophrenia.

## Ethics

The study has been approved by the Institutional Ethics Committee of ICMR-Centre for Ageing and Mental Health, Kolkata, India (18th IEC/CAM/1.1, dated 30.05.2023) and the Institutional Ethics Committee of All India Institute of Medical Sciences, Kalyani, India (IEC/AIIMS/Kalyani/certificate/2024/015, dated 31.01.2024). All participants will be recruited after obtaining written informed consent from themselves or their legally authorized representatives. The study will conform to the requirements of the Declaration of Helsinki, 1964; the Indian Good Clinical Practice guidelines; and the ICMR-National Ethical Guidelines for Biomedical and Health Research Involving Human Participants, 2017. Personal information about screened and enrolled participants will be collected, shared, and maintained to protect confidentiality before, during, and after the trial. A 2: 1 randomization technique will allow more patients to be enrolled in the PGx arm and benefit if the PGx-assisted treatment is favorable. The genotyping data will be revealed to the patients after 12 weeks. If PGx-assisted treatment is beneficial, it will be implemented for patients in Arm B (standard of care arm). All data source documents, including clinical reports and records necessary for the evaluation and reconstruction of the clinical trial, will be stored securely for five years, with a focus on ensuring the patient's confidentiality. The blood samples will be stored for future research in the systems pharmacology of antipsychotics.

## Protocol registration

The study has been registered with the Clinical Trials Registry–India (CTRI/2023/08/056210). The full protocol is available in the S2 File.

## Discussion

This study aims to set up an individualized preferred treatment prediction model through the genetic analysis of patients using different kinds of antipsychotics. It is important to ensure effective and safe management for patients with schizophrenia by improving the use of antipsychotic drugs. It has been found that patient characteristics and clinical circumstances could affect drug effectiveness; these patient factors are important in making treatment choices [49]. Treatment optimization using precision medicine is required to improve the efficacy and reduce the development of ADRs in patients on antipsychotic therapy. One of the main goals of precision medicine in such conditions is to use genetic information to improve safety, efficacy, and outcomes. PGx testing can be seen as a companion decision-support tool, under consideration of all relevant individual clinical and demographic information available.

To date, no randomized controlled trial has been conducted in patients with schizophrenia to evaluate the outcomes following treatment assisted by PGx using an extensive array of genes and antipsychotic drugs. Further, the cost-effectiveness analysis and the qualitative component (hybrid effectiveness-implementation design type 2 study) are the study novelties. The findings of this study will help in understanding whether PGx-assisted management (drug and dosage selection) supported by therapeutic drug monitoring has a clinical utility (safety and efficacy) or is

cost-effective in patients with schizophrenia. Another similar study is underway in China to assess the efficacy and adverse effects of antipsychotics and create a standard clinical cohort with a multi-dimensional index assessment (including PGx) of antipsychotic treatment for schizophrenia [50]. However, there is a limitation to this study. Since patients from the eastern Indian region will only be recruited, the generalizability of the study results is limited. Hence, a larger follow-up study is warranted in other populations. If the results are encouraging, PGx-assisted management could be incorporated into the standard treatment guideline for treating schizophrenia.

## Supporting information

**S1 File. SPIRIT (Standard Protocol Items: Recommendations for Interventional Trials) checklist.**
(PDF)

**S2 File. Full protocol.**
(DOCX)

## Acknowledgments

The authors gratefully acknowledge the feedback received in the protocol from Dr. Hisham Moosan, Scientist E (Medical), ICMR-National Institute of Implementation Research in Non-communicable Diseases, Jodhpur, India; Dr. Tamilarasu Kadhiravan, Professor, Department of Medicine, Jawaharlal Institute of Postgraduate Medical Education and Research, Puducherry, India; and Dr. Nabendu Sekhar Chatterjee, Head, Discovery Research Division (Extramural Projects), ICMR, New Delhi, India through a protocol development workshop conducted by the ICMR, New Delhi, India. The authors also acknowledge the scientific input received by the Technical Advisory Group, ICMR-National Taskforce on Safe and Rational Use of Medicines. Finally, the authors thank the Improving Use of Medicines team, Health System and Policy, Department of Global Public Health, Karolinska Institutet, Stockholm, Sweden for providing support.

## Author Contributions

**Conceptualization:** Aniruddha Basu, Atanu Kumar Dutta, Bhavani Shankara Bagepally, Saibal Das, Jerin Jose Cherian, Sudipto Roy, Pawan Kumar Maurya, Indranil Saha, Kumari Rina, Sucharita Mandal, Sukanto Sarkar, Asim Saha, Amit Chakrabarti.

**Funding acquisition:** Saibal Das.

**Methodology:** Manoj Kalita, Kalyan Bhowmik.

**Resources:** Kalyan Bhowmik.

**Software:** Manoj Kalita.

**Writing – original draft:** Aniruddha Basu, Atanu Kumar Dutta, Bhavani Shankara Bagepally, Saibal Das, Jerin Jose Cherian, Sudipto Roy, Pawan Kumar Maurya, Indranil Saha, Deepasree Sukumaran, Kumari Rina, Sucharita Mandal, Sukanto Sarkar, Asim Saha, Amit Chakrabarti.

**Writing – review & editing:** Aniruddha Basu, Atanu Kumar Dutta, Bhavani Shankara Bagepally, Saibal Das, Jerin Jose Cherian, Sudipto Roy, Pawan Kumar Maurya, Indranil Saha, Deepasree Sukumaran, Kumari Rina, Sucharita Mandal, Sukanto Sarkar, Asim Saha, Amit Chakrabarti.

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
