## [Decision Letter · Decision Letter 0]

3 Jan 2024

PONE-D-23-31935Pharmacogenomics-Assisted Schizophrenia Management: A Hybrid Type 2 Effectiveness-Implementation Study Protocol to Compare the Clinical Utility, Cost-Effectiveness, and BarriersPLOS ONE

Dear Dr. Das,

Thank you for submitting your manuscript to PLOS ONE. After careful consideration, we feel that it has merit but does not fully meet PLOS ONE’s publication criteria as it currently stands. Therefore, we invite you to submit a revised version of the manuscript that addresses the points raised during the review process.

**ACADEMIC EDITOR: **The reviewers have raised several comments on the statistical analysis. Authors are recommended to address the comments accordingly.==============================

We look forward to receiving your revised manuscript.

Kind regards,

Hoh Boon-Peng, PhD

Academic Editor

PLOS ONE

“This study is funded by ICMR, New Delhi, India through the ICMR-National Taskforce on Safe and Rational Use of Medicines (extra-mural fund).”

Additional Editor Comments:

The reviewers have raised some comments on the statistical analysis. Authors are suggested to address the comments accordingly before the manuscript can be accepted for publication.

Reviewers' comments:

Reviewer's Responses to Questions

**Comments to the Author**

1. Does the manuscript provide a valid rationale for the proposed study, with clearly identified and justified research questions?

Reviewer #1: Yes

Reviewer #2: Yes

2. Is the protocol technically sound and planned in a manner that will lead to a meaningful outcome and allow testing the stated hypotheses?

Reviewer #1: Yes

Reviewer #2: Partly

3. Is the methodology feasible and described in sufficient detail to allow the work to be replicable?

Reviewer #1: Yes

Reviewer #2: Yes

4. Have the authors described where all data underlying the findings will be made available when the study is complete?

Reviewer #1: Yes

Reviewer #2: Yes

5. Is the manuscript presented in an intelligible fashion and written in standard English?

Reviewer #1: Yes

Reviewer #2: Yes

6. Review Comments to the Author

You may also provide optional suggestions and comments to authors that they might find helpful in planning their study.

Reviewer #1: This manuscript is a study protocol for conducting a study in 3 phases: a randomized trial (phase 1), followed by a cost-effectiveness study (phase 2), and finally exploring a qualitative design (phase 3). The trial was approved by the respective Ethics/IRB board, and was registered under the Indian Clinical Trials Registry. The study objectives are on target. However, I mostly have some concerns/comments in the statistical design and analytical framework, and CONSORT guidelines, which may require attention:

1. Methods:

Methods reporting need some work. An orderly manner is suggested, following CONSORT guidelines, without repeating information, such as Trial Design, Participant Eligibility Criteria and settings, Interventions, Outcomes, sample size/power considerations, Interim analysis and stopping rules, Randomization (details on random number generation, allocation concealment, implementation), Blinding issues, etc, should be mentioned. The authors are advised to create "separate subsections" for each of the possible topics (whichever necessary), and that way produce a very clear writeup. They are advised to write it carefully, following nice examples in the manuscript below:

https://www.sciencedirect.com/science/article/pii/S0889540619300010

In the submitted version, there is an attempt in that regard in the right direction, and systematic presentation will enhance the appeal for readers.

Specific comments:

(a) For instance, the randomization and allocation concealment should be made very clear (they are NOT the same thing); the trial staff recruiting patients should NOT have the randomization list. Randomization should be prepared by the trial statistician, and he/she would not participate in the recruiting. Create a separate subsection!

(b) Sample size/power: The sample size/power statement presented (phase 1) does not provide the name of the test used, and the corresponding effect size the study was powered on.

The sample size statement for Phase 3 needs more details; it's not clear what 6-12 patients mean.

(c) Statistical analysis plan:

(c1) What's the plan under violations of the proportional hazards assumptions (phase 1) during analysis?

(c2) Mention the basis of the budget formula; is there any reference for considering it?

(c3) Some additional details regarding the proposed thematic analysis (phase 3) will be helpful for the readers.

2. Writing style:

Discussion Section: Given that the study will only enroll Indian nationals, the Discussion section should clearly allude to future studies on other populations and geographical regions to further validate the current findings. The findings from this protocol will only be limited to this population.

Reviewer #2: A well designed protocol. However, it would be better if authors can address the following comments:

1. Would be good to clarify what’s the meaning of “hybrid type 2 effectiveness-implementation research” in Introduction section.

2. Line 97: “The study duration is three years”. Since this is a study protocol paper, it should use future tense instead of present tense in the manuscript. Kindly amend the rest of the typo-mistakes throughout the manuscript.

3. Figure 1 is a table, rather than figure. Perhaps should renamed it.

4. Figure 1 (or table) needs more descriptive explanation since Line 104-106 didn’t provide enough description on whats the key message and rationale of the information provided. The figure (or table) should be self-explained as well so readers can apply the same study protocol in their future studies.

5. Line 108-126 should not be the exact wordings from Table 1. Ideally, authors should describe Table 1 in text and provides more information/justification/rationale to the readers.

6. Table 2, the rationale of selecting these genes should be clearly stated in the protocol.

7. Line 156: Perhaps more description on how the “computer generated random number” will work. What is the recommended software and how robust is the software?

8. Line 260: Kindly elaborate the rationale for this statement: “All future costs and consequences will be discounted at 3%.”

9. Line 297: “In-depth interviews and focused-group discussions of patients and their primary caregivers will be conducted” Would the patient/caregiver knows whether they are receiving PGx? Previously it was mentioned patient will be blinded in this study.If the patient is unaware of their arm, how would patient able to provide effective qualitative feedback?

10. PGx treatment protocol is not a new approach. It would be good to discuss the difference of this protocol vs other established protocols. Highlighting why certain amendment has been proposed or incorporated in this protocol in the Discussions section.

11. There are inconsistency in referencing format. Authors should relook into the formatting again.

7. PLOS authors have the option to publish the peer review history of their article (what does this mean?). If published, this will include your full peer review and any attached files.

Reviewer #1: No

Reviewer #2: No

---

## [Author Response · Author response to Decision Letter 0]

24 Jan 2024

Response to reviewers’ comments

We thank the editor and the reviewers for taking the time and effort to review the manuscript. We appreciate the valuable comments received; these will certainly improve the quality of the manuscript. We have responded to each comment below. 

Additional Editor’s Comments

1. The reviewers have raised some comments on the statistical analysis. Authors are suggested to address the comments accordingly before the manuscript can be accepted for publication.

Response: We have now modified the statistical analysis section.

Reviewer 1 Comments

1. Methods reporting need some work. An orderly manner is suggested, following CONSORT guidelines, without repeating information, such as Trial Design, Participant Eligibility Criteria and settings, Interventions, Outcomes, sample size/power considerations, Interim analysis and stopping rules, Randomization (details on random number generation, allocation concealment, implementation), Blinding issues, etc., should be mentioned. The authors are advised to create "separate subsections" for each of the possible topics (whichever necessary), and that way produce a very clear writeup. They are advised to write it carefully, following nice examples in the manuscript below: 

Comparative assessment of plaque removal and motivation between a manual toothbrush and an interactive power toothbrush in adolescents with fixed orthodontic appliances: A single-center, examiner-blind randomized controlled trial - ScienceDirect

In the submitted version, there is an attempt in that regard in the right direction, and systematic presentation will enhance the appeal for readers.

Response: We have followed the SPIRIT 2013 Statement (submitted as a supplementary file) that provides evidence-based recommendations for the minimum content of a clinical trial protocol. SPIRIT is widely endorsed as an international standard for trial protocols (Table 1 SPIRIT 2013 checklist: recommended items to address in a clinical trial protocol and related documents* (spirit-statement.org)). We have created the suggested sub-sections.

2. For instance, the randomization and allocation concealment should be made very clear (they are NOT the same thing); the trial staff recruiting patients should NOT have the randomization list. Randomization should be prepared by the trial statistician, and he/she would not participate in the recruiting. Create a separate subsection!

Response: Thank you for this suggestion. We have created a separate subsection on Randomization, blinding, and allocation concealment and described the details as follows:

Variable-sized block randomization will be performed based on computer-generated random numbers using an open-access software (sealed envelope™). Allocation concealment will be ensured using serially numbered opaque sealed envelopes. The randomization code and the envelopes for allocation concealment will be prepared by the study statistician having no role in the participant enrolment process. The treating psychiatrists will be enrolling the trial participants and will assign them to the treatment arms as per the allocation sequence but will not have the randomization list. The treating psychiatrists will be aware of the patient’s treatment arm (unblinded). The patients and the outcome assessors will not be revealed of the treatment arms and the genotyping results (patient and assessor blind) until the study completion (follow-up for 12 weeks).

3. Sample size/power: The sample size/power statement presented (phase 1) does not provide the name of the test used, and the corresponding effect size the study was powered on. The sample size statement for Phase 3 needs more details; it's not clear what 6-12 patients mean.

Response: For the Phase 1, we have mentioned the effect size:

Considering a difference of 0.6210 in the UKU-SERS score between the two arms, a standard deviation of 1.3410, a clinically meaningful difference of 1.18, a two-sided alpha error of 5%, a power of 90%, a randomization ratio of 2: 1, and a drop-out rate of 10%, the final estimated sample size is 249 (round-up to 250) [166 in Arm A (PGx arm) and 83 in Arm B (standard of care arm)]. The statistical tests are mentioned under Data management and statistical analyses section.

Patients, primary caregivers, and psychiatrists (six to twelve each) will be interviewed until saturation is obtained.39 This sample size is tentative and interview will continue till data saturation.

4. Statistical analysis plan:

What's the plan under violations of the proportional hazards assumptions (phase 1) during analysis?

Mention the basis of the budget formula; is there any reference for considering it?

Some additional details regarding the proposed thematic analysis (phase 3) will be helpful for the readers.

Response: Thank you for identifying this. For violation in the proportional hazards assumption in Cox's proportional hazards model, violations will be identified through graphical methods and a sensitivity analysis will be performed to evaluate the robustness of the results to different modeling assumptions. This might involve using different cutoffs for time intervals or assessing the impact of different modeling strategies. If assumptions are not violated, based on the statistical criteria, we will choose an appropriate model.

The reference to the budget formula is: Prinja S, Chugh Y, Rajsekar K, et al. National Methodological Guidelines to Conduct Budget Impact Analysis for Health Technology Assessment in India. Appl Health Econ Health Policy. 2021;19:811-823.

A thematic analysis will be performed. This will involve a meticulous process of familiarization with the data, coding segments into meaningful units, and iteratively organizing these codes into coherent themes that encapsulate key concepts or patterns. Because of the flexible yet rigorous approach, thematic analysis will enable the exploration of diverse perspectives, facilitating the extraction of rich, contextually embedded insights that deepen our understanding of the facilitators and barriers to implementing PGx-assisted management in patients with schizophrenia. 

5. Discussion Section: Given that the study will only enrol Indian nationals, the Discussion section should clearly allude to future studies on other populations and geographical regions to further validate the current findings. The findings from this protocol will only be limited to this population.

Response: We have mentioned this under the Discussion section. However, there is a limitation to this study. Since patients from eastern Indian region will only be recruited, the generalizability of the study results is limited.

Reviewer 2 Comments

1. A well-designed protocol. However, it would be better if authors can address the following comments:

Would be good to clarify what’s the meaning of “hybrid type 2 effectiveness-implementation research” in Introduction section.

Response: Thank you for this suggestion. We have added the following sentences in the Introduction:

Hybrid type 2 effectiveness-implementation research combines elements of both effectiveness and implementation research within a single study. This approach aims to not only assess the effectiveness of an intervention or program in real-world settings but also focuses on understanding and enhancing the implementation process.

2. Line 97: “The study duration is three years”. Since this is a study protocol paper, it should use future tense instead of present tense in the manuscript. Kindly amend the rest of the typo-mistakes throughout the manuscript.

Response: Thank you. We have made the necessary changes.

3. Figure 1 is a table, rather than figure. Perhaps should renamed it.

Response: I agree; however, this is as per the Author’s Guidelines of PLOS ONE (Submission Guidelines | PLOS ONE)

4. Figure 1 (or table) needs more descriptive explanation since Line 104-106 didn’t provide enough description on what’s the key message and rationale of the information provided. The figure (or table) should be self-explained as well so readers can apply the same study protocol in their future studies.

Response: We have updated Figure 1. Figure 1 depicts the schedule of assessment of various outcome measurements. The same is detailed under the Data collection and outcome measurements section. 

5. Line 108-126 should not be the exact wordings from Table 1. Ideally, authors should describe Table 1 in text and provides more information/justification/rationale to the readers.

Response: We have now described the eligibility criteria in more details.

6. Table 2, the rationale of selecting these genes should be clearly stated in the protocol.

Response: The genes and the specific single-nucleotide polymorphisms (SNPs) are selected based on their role in the pharmacokinetics and pharmacodynamics of the mentioned antipsychotics and their prevalence in the eastern Indian population.18,19

7. Line 156: Perhaps more description on how the “computer generated random number” will work. What is the recommended software and how robust is the software?

Response: Variable-sized block randomization will be performed based on computer-generated random numbers using an open-access software (sealed envelope™). This online program provides reliable randomization services to randomized controlled trials and has been used by many trial groups globally.

8. Line 260: Kindly elaborate the rationale for this statement: “All future costs and consequences will be discounted at 3%.”

Response: This implies that implies the application of a discount rate of 3% to account for the time value of money when considering future costs and consequences in the cost-effectiveness. Discounting is a common practice in pharmacoeconomics and it recognizes that the value of money tends to decrease over time due to factors like inflation, opportunity costs, and uncertainty.

9. Line 297: “In-depth interviews and focused-group discussions of patients and their primary caregivers will be conducted” Would the patient/caregiver knows whether they are receiving PGx? Previously it was mentioned patient will be blinded in this study. If the patient is unaware of their arm, how would patient able to provide effective qualitative feedback?

Response: Part 3 of the study will be performed chronologically after the completion of Part 1 and Part 2. The patients will then know their treatment arm. In fact, the genotyping data will be revealed to the patients after 12 weeks of treatment (Part 1). This is mentioned under the Ethics section.

10. PGx treatment protocol is not a new approach. It would be good to discuss the difference of this protocol vs other established protocols. Highlighting why certain amendment has been proposed or incorporated in this protocol in the Discussions section.

Response: Thank you. We have mentioned the following points in the Discussion section:

To date, no randomized controlled trial has been conducted in patients with schizophrenia to evaluate the outcomes following treatment assisted by PGx using an extensive array of genes and antipsychotic drugs. Further, the cost-effectiveness analysis and the qualitative component are the study novelties (hybrid effectiveness-implementation design type 2 study).

11. There is inconsistency in referencing format. Authors should relook into the formatting again.

Response: We have now corrected the reference formatting.

---

## [Decision Letter · Decision Letter 1]

29 Feb 2024

Pharmacogenomics-Assisted Schizophrenia Management: A Hybrid Type 2 Effectiveness-Implementation Study Protocol to Compare the Clinical Utility, Cost-Effectiveness, and Barriers

PONE-D-23-31935R1

Dear Dr. Das,

We’re pleased to inform you that your manuscript has been judged scientifically suitable for publication and will be formally accepted for publication once it meets all outstanding technical requirements.

Kind regards,

Hoh Boon-Peng, PhD

Academic Editor

PLOS ONE

Additional Editor Comments (optional):

Reviewers' comments:

Reviewer's Responses to Questions

**Comments to the Author**

1. Does the manuscript provide a valid rationale for the proposed study, with clearly identified and justified research questions?

Reviewer #1: Yes

Reviewer #2: Yes

2. Is the protocol technically sound and planned in a manner that will lead to a meaningful outcome and allow testing the stated hypotheses?

Reviewer #1: Yes

Reviewer #2: Yes

3. Is the methodology feasible and described in sufficient detail to allow the work to be replicable?

Reviewer #1: Yes

Reviewer #2: Yes

4. Have the authors described where all data underlying the findings will be made available when the study is complete?

Reviewer #1: Yes

Reviewer #2: Yes

5. Is the manuscript presented in an intelligible fashion and written in standard English?

Reviewer #1: Yes

Reviewer #2: Yes

6. Review Comments to the Author

You may also provide optional suggestions and comments to authors that they might find helpful in planning their study.

Reviewer #1: The authors were able to address my previous set of queries with a great degree of satisfaction. I have no further comments.

Reviewer #2: Authors addressed all concerns. it is recommended that this paper should be ready to be considered for publication.

7. PLOS authors have the option to publish the peer review history of their article (what does this mean?). If published, this will include your full peer review and any attached files.

Reviewer #1: No

Reviewer #2: No

---

## [Editor Report · Acceptance letter]

1 Apr 2024

PONE-D-23-31935R1 

PLOS ONE

Dear Dr. Das, 

I'm pleased to inform you that your manuscript has been deemed suitable for publication in PLOS ONE. Congratulations! Your manuscript is now being handed over to our production team.

Kind regards, 

on behalf of

Professor Dr Hoh Boon-Peng 

Academic Editor

PLOS ONE